# Hope Aspects of the Women’s Experience after Confirmation of a High-Risk Pregnancy Condition: A Systematic Scoping Review

**DOI:** 10.3390/healthcare10122477

**Published:** 2022-12-08

**Authors:** Mónica Antunes, Clara Roquette Viana, Zaida Charepe

**Affiliations:** Center for Interdisciplinary Research in Health, Institute of Health Sciences, Universidade Católica Portuguesa, 1649-023 Lisbon, Portugal

**Keywords:** high-risk pregnancy, hope, life experience, mental health, pregnancy complications, prenatal diagnosis, review

## Abstract

Background: Pregnancy is a period of transformation, hope, expectation, and worry for women and their families. A high-risk pregnancy refers to a pregnancy in which the mother and/or fetus are at greater-than-normal risk of complications, and it evokes a range of emotional and psychological experiences that largely depend on the care and support provided by health professionals. The purpose of this review is to summarize the existing literature on the lived experience of hope in women facing a high-risk pregnancy related to their own health and/or medical conditions related to the fetus. Methods: This review followed the Joanna Briggs Institute’s methodology. No limits on a date were applied to the search. Identified titles and abstracts were screened to select original reports and were cross-checked for any overlap of cases. We included studies that emphasized the experience of hope of pregnant women dealing with a pregnancy complication. Main Results: According to the results of the present scoping review, we found two main dimensions: women experiencing a high-risk pregnancy themselves and prenatal diagnosis. In both cases, the women were in a dilemma between hope and hopelessness. Conclusion: The findings demonstrate that women facing high-risk pregnancies struggle with multiple fears and concerns about their own health and the fetus’s health. Further research is needed to identify best practices for the care provided to the vulnerable populations.

## 1. Introduction

In 2017, approximately 810 women died from preventable causes related to pregnancy and childbirth [1]. Globally, more than 20 million women are at risk of high-risk pregnancies, which results in an estimated 830 deaths per day, more than 99% occurring in developing countries, and is more frequent among rural women and adolescents [2]. Nearly 22% of pregnant women develop a high-risk pregnancy [3]. As part of the sustainable development goals (SDG), countries have agreed on a new target to accelerate the decline of maternal mortality by 2030. The World Health Organization (WHO) considers high-risk pregnancies a major public health challenge, addressing the healthcare needs as a priority [1], and includes an ambitious target: decreasing the global maternal mortality rate to less than 70 per 100,000 births, with no country having a maternal mortality rate of more than twice that of the global average [1]. 

In most cases, the birth of a child is an experience filled with joy and happiness. However, when a mother is diagnosed with a medical condition or the child with a congenital anomaly, the parents’ experience can take on a different meaning [4]. A high-risk pregnancy is defined as any pregnancy in which there is a medical factor, maternal or fetal, that may potentially adversely affect the outcome of the pregnancy [5]. The most common maternal complications are gestational diabetes, preeclampsia and eclampsia, depression, sexually transmitted diseases, preterm labor, and placenta previa [3], while the most frequent congenital anomalies are heart malformations, neural tube defects, and Down syndrome. An anomaly may be genetic, infectious, or environmental in origin. In most cases, however, the cause is unknown, which makes it more difficult for parents to understand and accept the situation [6]. In many countries, congenital anomalies are important causes of perinatal morbidity and mortality, which can lead to chronic disabilities which may have severe consequences on individuals, families, healthcare systems, and societies [7]. The sudden sense of grief, loss, and guilt coupled with a fear of the unknown future often produces a great deal of anguish for parents [5], especially in pregnant women.

Every pregnancy is unique, physiologically, and a natural episode in a woman’s life, and all pregnant women experience physical, mental, and social changes in different manners [3]. The ways in which the changes inherent to this transition moment are integrated and experienced seem to be directly related to the woman’s personality, marital status, family, and social support [8]. 

Under normal conditions, pregnancy is a natural transition for women and their families. When a pregnant woman is diagnosed as being high-risk, she may find it difficult to cope with this new reality, leading to psychological and emotional consequences such as fear, guilt, shock, grief, frustration, worry, loneliness, and isolation [9]. A high-risk pregnancy with complications is one of the risk factors causing pregnant women to experience psychosomatic problems such as anxiety, depression, and distress, and to suffer impairments in their health [10]. A qualitative study also showed that besides medical problems, women experience behavioral, affective, and emotional problems. Moreover, they are also at risk of sociocultural and financial strains that often lead to feelings such as uncertainty, concern, and insecurity [11]. 

Women’s coping strategies during pregnancy demand multiple challenges in which hope and resilience play essential roles in managing stress and mental health. Hope is defined as the perceived ability to find pathways to desired goals and to motivate oneself to use those pathways [12]. As a focus of nursing practice, hope is defined as “(…) feelings of having possibilities, trust in others and in future, zest for life, expression of reasons and will to live, inner peace, optimism, associated with setting goals and mobilization of energy” [13] (p. 1). In a study published in 2019, the author defends that helping individuals and their family members to find meaning in suffering and striving to invoke a sense of constructive hope should be a fundamental aspect of health care [14]. Inspiring appropriate hope might be considered a concern to healthcare professionals whatever their specialty [14]. In this review, the lived experiences of women’s hope comprises spheres and dimensions [14]. According to Dufault and Martocchio, there are six dimensions associated with the concept of hope [15]: affiliative (relationships with women and God that can be expressed by individuals who seek or are receptive to others’ help in hope); affective (sensations and emotions that are part of the process of hope); cognitive and behavioral (interpret and judge in relation to hope and actions orientation towards the desired outcomes, respectively); contextual (life situations that surround, influence, and are a part of women’s hope); and temporal (focuses upon hoping within the women’s experience of time). 

Therefore, the aim of this review was to assess the state of knowledge regarding the lived experience of hope among women facing high-risk pregnancies that may endanger the health of the mother and/or fetus. This openness to the possibility of working hope in a perspective other than cure, and the commitment of nurses in the practice of promoting hope as a duty of care and a standard of good clinical practice, has led to the need to investigate the concept and look for new ways to better-inspire hope in women who face pregnancies with complications in the context of their health and/or that endanger the health of the fetus.

## 2. Materials and Methods

The protocol and proposed systematic review were drawn and conducted in accordance with the JBI methodology for systematic reviews [16]: define search strategy, study selection, assessment of methodological quality, data extraction, and data synthesis. The protocol was registered prospectively with the Open Science Framework on 26 May 2022 (https://osf.io/u9ns8 (accessed on 12 September 2022)). Registration DOI: 10.17605/OSF.IO/U9NS8.

### 2.1. Review Questions

The following question guided this scoping review: What are the hope aspects related to the women’s lived experience of a pregnancy that continues after confirmation of a high-risk pregnancy diagnosis? This question was divided into the following sub-questions: What type of evidence or study design exists in the area of women’s life experience of pregnancy research in relation to hope aspects after confirmation of a high-risk pregnancy diagnosis?; What aspects of hope in the life experience of pregnant women have been addressed during health care associated with a high-risk pregnancy?; What are the gaps in the nursing research in relation to hope interventions in the context of care for women with high-risk pregnancies? 

### 2.2. Inclusion Criteria

#### 2.2.1. Participants

Both the exploratory and textual components of this review considered qualitative and quantitative studies that include women’s lived experiences of pregnancy with a high-risk condition that may affect their own health or their baby’s health, including the prenatal diagnosis of a congenital anomaly, regardless of race, nationality, level of education, or religious affiliation. The study could also include any patient with an age above 18, primigravida and multigravida, and in the second or third trimester of their pregnancy. This review excluded any studies focusing on healthcare professionals and pregnant women who were healthy and did not have any risk factors for high-risk pregnancies.

#### 2.2.2. Concept

This review considered studies that explore woman’s hopeful experiences when proceeding with a high-risk pregnancy diagnosis, which may have included maternal and/or fetal complications. The included studies may or may not have directly addressed hope in their proposals having, however, addressed hope-related experiences and related concepts in their findings or women’s expectations. The aspects related to women’s experiences of hope included psychological well-being; uncertainty; finding a sense of normality; finding a new meaning to life experiences; setting realistic goals and objectives; imagining possibilities and seeking alternative solutions; an ability to share their pregnancies with other women and/or health care professionals; and expressing positive personal transformation and identifying positive psychological factors. The aspects related to women’s expectations will include difficulties in thinking about the child’s future; condition-related expectations (positive or negative outlook); and concerns about acceptance by family and social networks.

#### 2.2.3. Context

Women with high-risk pregnancies are usually referred to larger health centers for better treatment. In this review, women were assisted in high-risk maternal–fetal health consultations, which included the literature from any country or sociocultural setting.

#### 2.2.4. Types of Sources

This review considered quantitative, qualitative, and mixed study designs for inclusion. Systematic reviews and meta-analyses were also considered in this review.

### 2.3. Search Strategy 

The search strategy aimed to find studies published in the Portuguese, English, or Spanish languages, with no date limit. As recommended in all JBI types of reviews, a three-step search strategy was carried out [16].

An initial limited search took place on the EBSCO platform, where the databases CINAHL and MEDLINE (PubMed) were selected, as well as the Google Scholar platform. The search was performed using the keywords included in the PCC question. A second search with all identified keywords and index terms was undertaken across all of the included databases, using Boolean descriptors such as “OR” and “AND”. A search with all identified keywords and index terms was used to develop a full search strategy for PubMed (see Appendix A). In the third stage, reference lists of sources selected from the full text and/or included in the review were examined. The databases searched include CINAHL Complete (from EBSCO); Pubmed; Nursing and Allied Health Collection (from EBSCO); PsycINFO; Mediclatina (from EBSCO); and Scopus. 

### 2.4. Study/Source of Evidence Selection

Following the search, all identified citations were compiled and uploaded to Mendeley version 2.71.0 (Mendeley Ltd., Elsevier, Amsterdam, Netherlands), and duplicates were removed. After a pilot test, the titles and abstracts were screened by two or more independent reviewers for assessment against the inclusion criteria for the review. The resulting reference list was then uploaded to Rayyan (Qatar Computing Research Institute, Doha, Qatar). In the second phase of screening, the references of the selected studies were reviewed, and relevant studies were identified for a full review. The studies were classified into one of three categories: included, excluded, and uncertain. Later, the full texts of the “included” and “uncertain” studies were retrieved to identify potentially relevant studies, and their full citation details were imported into Rayyan.

The full texts of the selected citations were assessed in detail according to the inclusion criteria by two independent reviewers. Reasons for the exclusion of sources of evidence in full text that did not meet the inclusion criteria were recorded and reported in the scoping review. Any disagreements that arose between the reviewers at each stage of the selection process were resolved through discussion, or with an additional reviewer/s. The results of the search and the study-inclusion process are reported and presented in a preferred reporting items for systematic reviews and meta-analyses extension for scoping review (PRISMA-ScR) flow diagram [17]. 

### 2.5. Assessment of Methodological Quality 

In describing the quality of the selected articles (*n* = 15), the studies were appraised by all of the authors. Divergent views regarding the critical appraisal were reviewed until a consensus was achieved. The Hawker et al. [16] assessment tool, with a four-grade scale (1 = very poor; 2 = poor; 3 = fair; 4 = good) was used. The total scores ranged between 9 and 36, and higher scores indicated a higher quality. An article’s quality appraisal was centered on the following items: 1—abstract and title; 2—introduction and aims; 3—method and data; 4—sampling; 5—data analysis; 6—ethics and bias; 7—results; 8—transferability or generalizability; and 9—implications and usefulness.

### 2.6. Data Extraction

After the analyses of the title and abstract, duplicates and articles that did not correspond to the topic were excluded. Primary studies published in Portuguese, English, and Spanish were included, with no time limit. The search was conducted on 16 March 2022, with an update on 16 September 2022. A total of 479 articles were excluded, and another 4 full-text articles were excluded, two of them with an unsuitable concept and for the other two we did not receive a response from the authors to access the full text, leaving 15 for analysis (Figure 1). The titles and abstracts identified during the search were independently reviewed by the authors using the inclusion and exclusion criteria. The decision of whether to include or exclude studies was made by mutual agreement. 

### 2.7. Data Synthesis 

The 15 studies eligible for SR are described in Appendix B. The results are presented in narrative form. Considering the JBI guidelines [16], the synthesis of relevant data collected from each article was composed of the following elements: the identification of the article, hope and hopelessness experiences or expectations, aims, study design, study population/sample, context, population characteristics, typology, and main results.

## 3. Results

The generated demand resulted in 495 titles. After applying the inclusion/exclusion criteria and excluding duplicate studies, 15 studies were eligible. Of these, 14 were in the English language and 1 was in the Portuguese language (Brazil).

### 3.1. Characteristics of Sources of Evidence

The main characteristics of the fifteen articles were as follows: fifteen primary studies, four of which were conducted in Iran, two in the United States of America, one in Brazil, one in the United Kingdom, one in Belgium and the United Kingdom, one in Sweden, one in Australia, one in Paris, one in Africa, one in Malaysia, and one in Thailand. Only one of the studies found was quantitative (descriptive, prospective, and longitudinal); the rest of the studies were qualitative.

The selected studies were published between 2004 and 2021. The ages of the women ranged from 18 to 45 years. The sample sizes varied from nine to seventy-two women who used the services of specialized centers, hospitals, or clinics for the specific medical condition presented.

There are studies (see Appendix B) that have focused on women’s experiences with maternal medical conditions [18,19,20,21,22,23,24,25,26,27], and other studies that have focused on the prenatal diagnosis of fetal pathology [28,29,30,31,32]. 

Few studies have addressed the aspects of hope during health care associated with high-risk pregnancies, and only one study was quantitative, while fourteen studies were qualitative.

The empirical studies described the lived experiences of women during pregnancy in the second and third trimesters with varied medical conditions. 

Authors such as Behboodi-Moghadam, Khalajinia, Nasrabadi, Mohraz and Gharacheh (2016), and Sanders (2008) explored the experiences of women diagnosed with HIV at each stage of pregnancy [18,20,23]. This same medical condition was the focus of other studies that examined these experiences in the prenatal period in South Africa [25], Thailand [24], and Brazil [27]. On the other hand, authors such as Tong, Brown, Winkelmayer, and Craig and Jesudason described the beliefs, values, and experiences of pregnancy in Australian women with CKD to inform on pre-pregnancy counseling and pregnancy care [21].

The experiences and perceptions of women with high-risk pregnancies are focused on topics regarding health and care practice issues/needs [22,30]. Tayeh, Jouannic, Mansour, and Kesrouani and Attieh explored patients’ perceptions of the prenatal diagnosis of fetal cardiac pathology and their reasons for deciding to continue the pregnancy despite being eligible for the medical termination of their pregnancy [30]. Norhayati, Hazlina, Hussain, Asrenee and Sulaiman examined women’s experiences of near-misses and their perceptions of quality of care in a retrospective study [26].

A theoretical framework for the process of adaptation following a fetal anomaly diagnosis was provided based on the proposal in the study of Lalor, Begley and Galavan [28]. In this study, data was collected from Irish women’s experiences of carrying a baby with a fetal abnormality to the end and beyond birth [28,29]. Integrated into the fetal abnormality, other authors describe the women’s reactions to the discovery of fetal hydronephrosis in the context of uncertainty about prognosis [19] and the women’s experiences during pregnancy with a child with a known, nonlethal congenital abnormality [32]. 

The concepts and strategies that women in the UK and Belgium use when considering maternal–fetal surgery as an option for the management of spina bifida in their fetus, and how this determines the acceptability of the intervention, were researched in [31].

### 3.2. Critical Appraisal within Sources of Evidence 

The study’s quality was high. Methods, ethics, and bias, as well as transferability, were the studies’ main limitations, and the critical appraisals ranged from 32 to 36 (see Appendix C).

### 3.3. Results of Sources of Evidence

According to the results of the present systematic review, women who experience a high-risk pregnancy or a prenatal diagnosis of fetal abnormality found themselves in a dilemma between hope and hopelessness and, in some cases, a second dilemma between terminating or continuing the pregnancy. 

To answer the question: “What are the hope aspects related to the women’s life experience of a pregnancy that continues after confirmation of a high-risk pregnancy diagnosis?”, we were able to find two subcategories within the main category of “Hope experiences in women with a high-risk pregnancy.” We can distinguish hopelessness and hope experiences. In nine studies it was possible to analyze that the risk involved was related to the women themselves in the context of their medical conditions (see Table 1). 

The studies we found related well to the hope and hopelessness women experience in the context of prenatal diagnoses that could endanger the fetus’s health (see Table 2). 

Some studies have established a link between women’s experiences of hope and the positive outcomes associated with this perception. The following results in women with medical conditions include the change from their experiences of uncertainty to new solutions and/or possibilities about a future uncertainty. These women focus their hope on the child and the privilege of experiencing pregnancy. 

On the other hand, studies conducted with women with prenatal diagnoses are focused on outcomes that are associated with a positive future for the child and the realization that they made the right decision when they decided not to terminate the pregnancy. 

The hope experiences related to women have more expression in the group of women who have the prenatal diagnosis without medical conditions in themselves. However, after a prenatal diagnosis of a fetal anomaly in the initial phase, some women hoped that they would hear that the diagnosis was a mistake [28]. 

Each person has different coping strategies when faced with stress [19]. Women should be clearly informed about the options and decisions they may need to make in cases of abnormal tests and prenatal screenings [29,32].

## 4. Discussion

In the study of high-risk pregnancies with different medical problems, hopelessness experiences are implicitly related to pregnancy worries, concerns about the child, future pregnancy, relationships and support with others, and higher costs [18]. 

When the risk was associated with the diagnosis of HIV during pregnancy, the attribute that emerged in all studies was fear of the cruelty of stigma, stereotyping, discrimination, and judgment [20,23,24,25,27]. 

Studies also showed the existence of concerns about the transmission of the virus to the baby and possible effects on their health [20]; emotional distress, ambivalence about pregnancy and motherhood [23], an association with negative self-image, loneliness, feelings of isolation, fears of loved ones, blame, and that they avoided any romantic relationships, and fears of being sick, going to the hospital, or dying [25]. In the same population, Ross et al. found that women perceived their lives as a struggle [24], such as with shock, fear, anxiety, and depression; with sharing one’s struggling with others and that they struggle to care for their baby, especially after birth; and that they struggle through ups and downs. The consciousness of fragility, noxious self, denied motherhood, social jealousy, and fear of genetic transmission have been described in the study of women with chronic kidney disease [21]. 

Studies focused on maternal near-miss experiences showed many fears and concerns [22,26]: fear of being unable to become pregnant again, fear of raising their child without siblings, fear of carrying about their child without a mother, fear of remarriage of a spouse if the spouse wanted more children, fear of becoming pregnant again and experiencing postpartum complications, fear of not being able to adjust to the complications and grieving for a long time, and guilt, intolerance of pain, irritability, and postpartum depression.

In the optical of mental health, other emotions have an expression in this analysis, such as anxiety, discouragement, and numbness, fear for their own life and the life of their baby, feelings of death, and feelings of an incomplete self because they are a woman without a uterus and a baby [26].

In a study that looked at the experiences of women with various pathologies during pregnancy, hope was found to be related to the adaptation to challenges, the belief that conditions will improve, and being hopeful about the future [18]. 

Children were seen as a “divine gift”, a chance to correct past mistakes, and to be good, loving mothers [20,23,27]. 

Religiousness resurfaced with the wish for a child to give meaning to their life’s and to help with the construction of a female identity, because to be a mother was stronger than any problem [27]. Hope was also identified in protecting children from contracting the disease and from stigma in the case of HIV disease [23,24,25].

In general, the studies reported that a positive pregnancy experiences was an important source of support and hope for women [27]. By participating in spiritual practices, women believed that God would respond to their needs and take care of their children [20,25,27]. Spirituality and resorting to God and Imams are the most common attributes of hope found in the found studies [18,22,25,26,27].

Other forms of adaptations included the natural maternal disposition to focus on their child, which was a motivator to seek treatment and a source of strength to continue living [21,26].

When a prenatal diagnosis occurs, studies identified the following women’s hopelessness attributes: the sense of disbelief, stress, doubting the struggle, shock, anger, and fear of developing a bond with the baby who may die [19,28,29,32]; anxiety and fears about the unknown [19]; guilt by the loss of the perfect baby [32].

Only one study analyzed the hopeless experience in the case of a woman who had maternal–fetal surgery [31]. Emotions such as uncertainty about their future child’s quality of life, fear of potential complications of the surgery and the possibility of losing their unborn child, fear of not waking up after the operation, and post-traumatic stress and depression were related [31].

In the rebuilding phase after prenatal diagnosis, a positive vision of the future seems to develop (whatever that may be) as the woman processes her experiences in such a way as to reconstruct the future and adjust her earlier beliefs about pregnancy and the world in general [28]. For example, in the context of fetal surgery, women felt strong feelings of responsibility and determination to do anything to improve their future child’s health outcomes [31].

The study conducted by Oscarsson et al. found that women’s experiences of hope were based on going through a crisis and knowing that they were doing the right thing [19]. Irani et al. compared the emotional experiences of women that decided to continue or terminate the pregnancy after the prenatal diagnosis of fetal anomalies [29]. The results identified in this population include a dilemma between hope and worry maintained by a positive attitude toward childbirth to cope with the situation and/or a return to normality [29]. In another study that compared the experiences between continuing and terminating the pregnancy, the authors cited religious beliefs and convictions and the belief that the baby would survive after birth [30]. Other sources of hope were found when there were good sources of information, time to prepare, support from family and friends, spiritual beliefs, staying busy with work and other activities and empathizing with the baby [32].

## 5. Conclusions and Implications for Practice

This review demonstrated the meanings of the lived experiences of women with high-risk pregnancies due to maternal medical conditions or prenatal diagnoses. 

The hope aspects related to the women’s lived experiences of pregnancies that continued after the confirmation of a high-risk pregnancy diagnosis were addressed during health care and were analyzed with a personal woman’s attributes.

The type of evidence or study design that exists in women’s life experiences of pregnancy research in relation to hope aspects after the confirmation of a high-risk pregnancy diagnosis is mostly qualitative. In the studies analyzed, the qualitative design stood out in terms of methodology (14 of the 15 studies analyzed in this review). This aspect is related to the type of the main review question, as we intended to understand the experiences of hope and not measure them quantitatively.

In the studies that explored high-risk pregnancies, the main issues identified were women’s fears for their own lives and the lives of their babies, the impossibility of getting pregnant again, postpartum complications, and an incomplete self. On the other hand, women facing a prenatal diagnosis of fetal anomalies faced the fear of developing an attachment to a baby who may die, the loss of the perfect baby, and the fear of possible complications of treatment. In both cases, women went through phases with feelings of emotional distress, ambivalence, disbelief, stress, struggle, anxiety, and depression.

Most women facing a high-risk pregnancy used religion, spirituality, and faith in God as coping mechanisms. Some others relied on good sources of information, time to prepare, support from family and friends, employment at work, and empathy with the baby. These women had a strong sense of responsibility for the treatment and saw it as an opportunity that they had been given. If they decided to continue with the pregnancy, they focused on the idea of doing the right thing. They hoped for the best possible outcome and that everything would be okay in the end.

In relation to hope interventions in the context of care for women with high-risk pregnancies, the gaps in the nursing research are about hope interventions. The results of different studies describe an implicit proposal to develop research about hope interventions in the context of care for women with high-risk pregnancies. However, we believe that women who are well informed about their situation and treatment are most likely to adapt and comply with treatment. In addition, recognizing the benefits of religious faith in situations of uncertainty is important and helps women to adjust to challenging situations. 

It is extremely important to monitor women’s emotional and psychological reactions after a prenatal anomaly diagnosis, not only throughout the pregnancy but also in the postnatal stages. Nurses and midwives have a privileged position in relation to women and can help them overcome difficult challenges in the present and future.

To establish a comparison between women’s responses in the different conditions (maternal medical situations and prenatal diagnoses), it is highly important to increase our understanding of the impacts of these experiences in this population.

## 6. Limitations 

The present scoping review was based mainly on qualitative studies conducted on a limited number of mothers with high-risk pregnancies, so the results cannot be generalized to similar populations. 

## Figures and Tables

**Figure 1 healthcare-10-02477-f001:**
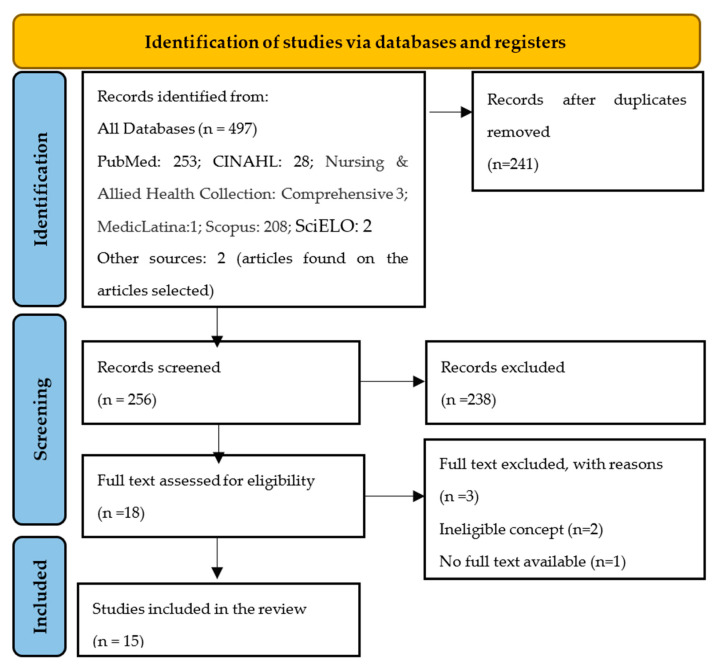
Flowchart of the selection and screening process of the systematic review articles according to the PRISMA method.

**Table 1 healthcare-10-02477-t001:** Hopelessness and hope experiences in women with medical conditions.

Hopelessness Experiences	Hope Experiences
- Worries and concerns about the child, future pregnancy, relationships, and higher costs [18] - Stigma, stereotyping, discrimination, and judgment [20,23,24,25,27] - Emotional distress and ambivalence [23] - Negative self-image, loneliness, feelings of isolation, and blame [25] - Fears for their own life and the life of their baby, unable to become pregnant again, and postpartum complications [26] - Struggle, shock, anxiety, and depression [24] - Conscious of fragility and fear of genetic transmission [21] - Not being able to adjust to the complications, grieving for a long time, guilt, intolerance to pain, and irritability [22,26]	- Adaptation to challenges [18] - Belief that conditions will improve [18,27] - Positive about the future [18,25,28] - Children as a “divine gift”, a chance to correct past mistakes, to be good, a loving mother [20,23] - Children as a meaning to their life and a motivator that helps to face challenges [24,27,28] - Pregnancy as construction of the female identity [27] - Medicines as hope, protecting their own life and protecting an unborn child [24,25] - Positive pregnancy experiences as a source of hope [21,24] - Religion, spirituality, faith in God [18,20,22,26,27] - Valuing life, gratitude, and focus on what is good [21]

**Table 2 healthcare-10-02477-t002:** Hopelessness and hope experiences in women with prenatal diagnoses.

Hopelessness Experiences	Hope Experiences
- Disbelief, stress, and struggle [28] - Fear of developing a bond with a baby who may die [28] - Anxiety and fears about the unknown, worry, stress, and depression [19,28,29,32] - Feelings of grief, shock, anger, panic, distress, and guilt [29,32] - Loss of perfect baby and fear of potential complications of the treatment [31]	- Positive vision of the future [28,31] - Reconstruct the future and adjust earlier beliefs [28,31] - Strong feelings of responsibility about the treatment, seen as an opportunity that was given [31] - Determination to do anything that would improve their future child’s health outcomes [19] - The sense of doing the right thing by continuing the pregnancy [29] - Hope for the best possible outcome and that everything would be all right at the end [29] - Positive attitude toward childbirth [29,30,31] - Hope to return to normal [30] - Other sources of hope, such as religious and spiritual beliefs, good sources of information, time to prepare, support from family and friends, staying busy at work, and empathizing with the baby [30]

## Data Availability

Not applicable.

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
