# Peer review of "Hope Aspects of the Women’s Experience after Confirmation of a High-Risk Pregnancy Condition: A Systematic Scoping Review"

_healthcare, 2022, doi:10.3390/healthcare10122477_

Round 1
Reviewer 1 Report
The article focus on hope, an important aspect for coping and resilience in the context of high risk pregnancy. Introducition could be more sinthetic. I suggest to take away the second paragraph on page 2, and to find another bibliography to mention malformations, genetic and congenital diseases (page 2 lines 46 – 49, it is not recommended to cite a citation ). The sentece on page 2, lines 72-74 needs revision. The items ‘review questions’ and ‘inclusion criteria’, in my opinion, are parts of ‘materials and methods’.How many articles were selected for the review? 15? Or 13 (line 242)? Or 14 (line 245)?The results densify evidence and point to advances in practices, but I found it too long, a little exhausting to read.
Reviewer 2 Report
Reviewer comments
The present review paper describes aspects of hope in women experiencing high-risk pregnancy. Both maternal conditions and fetal abnormalities have been included and hope/hopelessness have been evaluated. Hope has become a fundamental aspect of healthcare contributing to better outcome and mental health of patients. This paper can be of interest because of detailed presentation of different aspects of hope/hopelessness and questions that pregnant women struggle with. However, there are some recommendations to be addressed for the manuscript:
- The authors mentioned congenital anomalies as a cause of high-risk pregnancy; what about other causes of high-risk pregnancy such as gestational diabetes or preeclampsia? Gestational diabetes and preeclampsia are quite frequent complications of pregnancy and I suggest to present hope experiences of women having these complications.
- Did the authors consider the studies where the connection between level of hope/positive emotions and positive outcome has been determined? If there are such references, I suggest to emphasize them.
- There are review questions presented in the manuscript. I think that the responses to these questions should be clearly stated in conclusion.
- In review concept it has been stated that “studies explore women’s hopeful experiences” and that these were qualitative studies, but are there studies using some quantitative measurement of these experiences? Then, it may be useful to mention studies (if there are such studies) that quantify women who had hope and women who were hopeless and compare these frequencies.
- Did the authors have an insight into possible changes in hope/hopelessness experience among some women? For example, some women with hope went to hopelessness during pregnancy and vice versa.
- There is a discrepancy between number of studies included in review – (15 studies in chart = 15; 14 are eligible = 14; 12 in English and 1 in Portuguese = 13; 15 studies in Word file = 15). Please, correct these numbers.
- Sentence “Women experiencing high-risk pregnancy themselves and prenatal diagnosis” (line 19-20) – please better specify that these conditions all refer to high-risk pregnancy, but in one case these were mother’s conditions and diseases and in another case it was condition related to fetus.
- Sentence “According to Platt et al cited by Isaacs and Andipatin…” (line 46) – I think that the it is enough to choose one reference, not necessary to mention two.
